



# Modeling Trans-Pacific Transport and Stratospheric Intrusion of Tropospheric Ozone using Hemispheric CMAQ during April 2010: Part 2. Examination of Emission Impacts based on the Higher-order Decoupled Direct Method

Syuichi Itahashi[1], Rohit Mathur[2], Christian Hogrefe[2], Sergey L. Napelenok[2], and Yang Zhang[3]

[1] Environmental Science Research Laboratory, Central Research Institute of Electric Power Industry (CRIEPI), 1646 Abiko, Abiko, Chiba 270–1194, Japan
[2] Environmental Protection Agency (EPA), Computational Exposure Division, National Exposure Research Laboratory, Office of Research and Development, Research Triangle Park, NC 27711, USA
[3] Department of Marine, Earth, and Atmospheric Sciences (MEAS), North Carolina State University (NCSU), Campus Box 8208, Raleigh, NC 27695, USA

*Correspondence to*: Syuichi Itahashi (isyuichi@criepi.denken.or.jp)

**Abstract.**

The state-of-the-science Community Multiscale Air Quality (CMAQ) Modeling System which has recently been extended for
hemispheric-scale modeling applications (referred to as H-CMAQ), is applied to study the trans-Pacific transport, a phenomenon recognized as a potential source of air pollution in the U.S.A., during April 2010. The results of this analysis are presented in two parts. In the previous part 1 paper, model evaluation for tropospheric ozone ($O_3$) was presented and an air mass characterization method was developed. Results from applying this newly established method pointed to the importance of emissions as the factor to enhance surface $O_3$ mixing ratio over the U.S.A. In this subsequent part 2 paper, emission impacts
are examined based on mathematically rigorous sensitivity analysis using the higher-order decoupled direct method (HDDM) implemented in H-CMAQ. The HDDM sensitivity coefficients indicate the presence of a $NO_x$-sensitive regime during April 2010 over most of the northern hemisphere. By defining emission source regions over the U.S.A. and East Asia, impacts from these emission sources are examined. At the surface during April 2010, the emission impacts of the U.S.A. and East Asia are comparable over the western U.S.A. with a magnitude of about 3 ppbv impacts on a monthly mean of all hours basis whereas
the impact of domestic emissions dominates over the eastern U.S.A. with a magnitude of about 10 ppbv impacts on a monthly mean basis. The positive correlation (r=0.63) between surface $O_3$ mixing ratios and domestic emission impacts is confirmed. In contrast, the relationship between surface $O_3$ mixing ratios and emission impacts from East Asia exhibits a flat slope when considering the entire U.S.A. However, this relationship has strong regional differences between the western and eastern U.S.A.; the western region exhibits a positive correlation (r=0.36-0.38) whereas the latter exhibits a flat slope (r<0.1). Based
on the comprehensive evaluation of H-CMAQ, we extend the sensitivity analysis for $O_3$ aloft. The results reveal the significant impacts of emissions from East Asia on the free troposphere (defined as 750 to 250 hPa) over the U.S.A. (impacts of more than 5 ppbv), and the dominance of stratospheric intrusions on upper model layer (defined as 250 to 50 hPa) over the U.S.A.





(impacts greater than 10 ppbv). Finally, we estimate changes of trans-Pacific transport by taking into account recent emission trends from 2010 to 2015 assuming the same meteorological condition. The analysis suggests that the impact of recent emission changes on changes in the contribution of trans-Pacific transport to U.S.A. O₃ levels was insignificant at the surface level and was small (less than 1 ppbv) over the free troposphere.

## 5  1 Introduction

Tropospheric ozone (O₃) is a secondary air pollutant produced through photochemical reactions including nitrogen oxides (NOₓ) and volatile organic compounds (VOCs) (Haagen-Smit and Fox, 1954). Tropospheric O₃ plays an important role by producing hydroxyl radicals (OH) which control the oxidizing capacity (Logan, 1985). O₃ at the surface level poses significant human health impacts; hence many countries have an air quality standard for its ambient mixing ratios. The National

Ambient Air Quality Standard (NAAQS) of O₃ in the U.S.A. is set on the annual 4[th] highest maximum daily 8-h concentration (MD8O3) averaged over three years. Its threshold value was set at 70 ppbv in 2015 (EPA, 2018). An analysis of trends in surface O₃ observation during the period of 1998 and 2013 in the U.S.A. indicated that the highest O₃ mixing ratio have been decreasing responsive to reductions in O₃ precursor emissions (Simon et al., 2015). Regarding O₃ pollution in the U.S.A., sources enhancing O₃ mixing ratios are not limited to national emissions. One issue of potential concern is the dramatic

variation of anthropogenic emissions in East Asia which has been recognized as an important source for the U.S.A. through previous research on trans-Pacific transport (e.g., Jacob et al., 1999; Fiore et al., 2002; Wang et al., 2009, 2012; Lin et al., 2012a; Huang et al., 2017; Guo et al., 2018; Jaffe et al., 2018). Stratosphere-to-troposphere transport (STT) is another process affecting tropospheric O₃ pollution (Lelieveld and Dentener, 2000). The fraction of stratospheric origin on tropospheric O₃ varies by location and season, is strongly dependent on the tropopause altitudes and is an active research area (e.g., Fiore et

al., 2003; Lin et al., 2012b; Mathur et al., 2017). Literature estimates of the contributions of these two factors are summarized in the part 1 paper (see, Table 1 of Itahashi et al., 2019). The occurrence of these trans-Pacific transport and stratospheric intrusion can be related to the mid-latitude jet stream, and this is controlled by La Niña and El Niño. The springtime trans-Pacific transport may be enhanced following an El Niño winter due to the eastward extension of the atmospheric circulation over the Pacific-North America sector and the southward shift of the subtropical jet stream. The stratospheric intrusions may

be enhanced following a La Niña winter due to a meandering of the jet stream (Lin et al., 2015). Because enhancement of trans-Pacific transport is expected after the 2009-2010 El Niño winter, April 2010 is selected as the study period in the current analysis.

As illustrated in the part 1 paper, the objective of this sequential research is to better understand the relative contributions of precursor emissions from the U.S.A. and East Asia and also the impacts of STT on air quality in the U.S.A.

during spring time. To quantify these contributions, we used the model of Community Multiscale Air Quality (CMAQ) version 5.2 applied for hemispheric-scale analysis (H-CMAQ) (Mathur et al., 2017). The current study extends our previous analysis



(Itahashi et al., 2019; hereafter referred to as Part 1). A brief summary of the findings from that analysis and the motivation for this study is presented subsequently.

## 2 Summary of Part 1 and Motivation for Part 2

5       The model of H-CMAQ was configured with a horizontal grid spacing of 108 km with 187×187 grids to cover the entire Northern Hemisphere on 44 terrain-following vertical layers from the surface to 50 hPa (Mathur et al., 2017). The emission inputs are based on the modeling experiments of Hemispheric Transport of Air Pollution version 2 (HTAP2), and the description of this emission dataset can be found in relevant studies (Janssens-Maenhout et al., 2015; Pouliot et al., 2015; Galmarini et al., 2017; Hogrefe et al., 2018). For gas-phase and aerosol chemistry representation, cb05e51 and aero6 with

nonvolatile primary organic aerosol (POA) were used, respectively (Simon and Bhave, 2012; Appel et al., 2017), and further included a condensed representation of halogen chemistry which relate to $O_3$ loss in maritime environments (Sarwar et al., 2015). In terms of the stratospheric $O_3$ behavior, a robust indicator to distinguish between stratospheric and tropospheric air masses is potential vorticity (PV). A value of 2 PVU (1 PVU = $10^{-6}$ $m^2$ K $kg^{-1}$ $s^{-1}$) is suggested as the identification of stratospheric air (e.g., Hoskins et al., 1985). $O_3$ mixing ratios and PV are correlated, and $O_3$/PV ratios are used in H-CMAQ to

specify the model top $O_3$ mixing ratio. Starting with H-CMAQ version 5.2, a dynamic $O_3$/PV function has been implemented to account for the seasonal, latitudinal, and altitude dependencies of this relationship (Xing et al., 2016). The H-CMAQ simulation in this study started from 1 March 2010 and was initialized by three-dimensional chemical fields from prior model simulations for 2010 described in Hogrefe et al. (2018); March was discarded as a spin-up period and April was selected as analysis period.

20       To evaluate the performance of H-CMAQ simulations, the part 1 paper computed the Pearson's correlation coefficient (R) with student's $t$-test for the statistical significance level, the normalized mean bias (NMB), and the normalized mean error (NME). The analysis of ground-based mixing ratios included observations at 52 sites of the World Data Centre for Greenhouse Gases (WDCGG) over the northern hemisphere (WDCGG, 2018), 9 sites of the Acid Deposition Monitoring Network in East Asia (EANET) over Japan (EANET, 2018), and 81 sites of the Clean Air Status and Trends Network (CASTNET) over the

U.S.A. (CASTNET, 2018). Based on more than 4000 observation-model pairs of MD8O3, the results of this analysis showed good model performance with R around 0.5-0.6, NMBs around −10%, and NMEs around 10-20%. In addition to this ground-based analysis, vertical $O_3$ profiles were evaluated for three vertical layer ranges: from surface to approximately 750 hPa (i.e., boundary layer), approximately 750-250 hPa (i.e., free troposphere), and approximately 250-50 hPa (i.e., upper model layers) following the previous work of Hogrefe et al. (2018). Comparisons of vertical $O_3$ profile with ozonesonde observations

revealed that H-CMAQ can capture $O_3$ behavior well over the boundary layer. However, systematic underestimations by H-CMAQ over free troposphere were found with NMBs up to −30%, especially during strong STT events. Comparisons of modeled tropospheric $O_3$ columns with observed satellite data (NASA, 2018) indicate that H-CMAQ can generally capture the



northern hemispheric tropospheric $O_3$ column distributions with lower column amounts over the Pacific Ocean near the equator and higher column amounts over the mid-latitudes.

For the estimation of STT, a air mass characterization technique was newly developed. This was derived based on the ratio of modeled $O_3$ mixing ratios and an those of inert tracer for stratospheric $O_3$ to judge the relative importance of photochemistry and then determine whether an air mass is of stratospheric origin if the photochemistry is weak. The estimated STT showed day-to-day variations both in the impact magnitude and the air mass origin. The relationship between surface $O_3$ levels and estimated stratospheric air mass in the troposphere showed a negative slope, indicating that high surface $O_3$ mixing ratios at most locations were driven by other factors (e.g., emissions). In contrast, the relationship at elevated sites exhibits a slight positive slope, indicating a steady STT contribution to $O_3$ levels.

Because high surface $O_3$ mixing ratios were determined to be caused by emissions, this subsequent part 2 paper focuses on the analysis of emission impacts from the U.S.A. and East Asia. To examine these emission impacts, the traditional brute force method (BFM) approach of varying input parameters (e.g., emission) one-at-a-time is frequently used (e.g., Clappier et al., 2017). The application of the decoupled direct method (DDM) in H-CMAQ has been initiated to investigate the trends of $O_3$ distribution (Mathur et al., 2018a). In this study, we use the higher-order decoupled direct method (HDDM) implemented in H-CMAQ, which enables accurate and computationally efficient calculations of the sensitivity coefficients required for evaluation of the impact of input parameters variations on output chemical concentrations (Hakami et al., 2003; Cohan et al., 2005; Napelenok et al., 2008; Kim et al., 2009; Napelenok et al., 2011; Itahashi et al., 2013; Itahashi et al., 2015). The manuscript is organized as follows. The HDDM is described in Section 3. Analysis of $O_3$ sensitivity regimes over the entire northern hemisphere is presented in Section 4.1. By defining source regions over the U.S.A. and East Asia, the impacts of emissions from these regions on surface level $O_3$ over the U.S.A. are examined in Section 4.2. We then extend the analysis to $O_3$ aloft and present the results in Section 4.3. Trans-Pacific transport may have changed due to recent emission changes in East Asia, and the effects of these changes are estimated by considering the emission changes after 2010. This is discussed in Section 4.4. Finally, Section 5 summarizes the conclusions of our sequential papers.

## 3 Description of HDDM

Response of chemical concentrations to perturbations in model parameters (e.g., emissions, initial condition, boundary condition, reaction rate constants, etc.) can be investigated through sensitivity analysis. A perturbed sensitivity parameter, $p_i$, has the following relationship with the unperturbed sensitivity parameter, $P_i$, in the base-case simulation:

$$p_i = \varepsilon_i P_i = (1 + \Delta \varepsilon_i) P_i \qquad (1)$$

where $\varepsilon_i$ is a scaling factor with a nominal value of 1, and $\Delta\varepsilon_i$ is a perturbed scaling factor (e.g., $\varepsilon_i$ is 0 and then $\Delta\varepsilon_i$ is −1 for zero emission simulation). Here, the response of a chemical concentration, $C$, against the perturbations in a sensitivity





parameter, $p_i$, is defined as sensitivity coefficients, $S_i$. The semi-normalized first- and second-order sensitivity coefficients, $S_i^{(1)}$ and $S_{i,j}^{(2)}$ are defined as follows:

$$S_i^{(1)} = P_i \frac{\partial C}{\partial p_i} = P_i \frac{\partial C}{\partial (\varepsilon_i P_i)} = \frac{\partial C}{\partial \varepsilon_i} \qquad (2)$$

$$S_{i,j}^{(2)} = P_i \frac{\partial C}{\partial p_i} P_j \frac{\partial C}{\partial p_j} = P_i \frac{\partial C}{\partial (\varepsilon_i P_i)} P_j \frac{\partial C}{\partial (\varepsilon_j P_j)} = \frac{\partial^2 C}{\partial \varepsilon_i \partial \varepsilon_j} \qquad (3)$$

Because $\varepsilon_i$ and $\varepsilon_j$ are unitless, $S_i^{(1)}$ and $S_{i,j}^{(2)}$ have the same units with the chemical concentration, $C$. Physically, $S_i^{(1)}$ represents the impact of one variable $p_i$ on the concentration, $C$, and $S_{i,j}^{(2)}$ measures how a first-order sensitivity of $S_i^{(1)}$ changes under the changes of another variable $p_j$, and can be used to explore the nonlinearities in a system. When i=j, $S_{i,i}^{(2)}$ represents the local curvature of the relationships between concentration and one parameter. HDDM calculates semi-normalized first- and second-order sensitivity coefficients simultaneously in a single model simulation based on a governing set of sensitivity equations which have a formulation analogous to the atmospheric species equations in the CMAQ modeling system.

To project the fractional perturbation from the base-case simulation, the corresponding concentration can be approximated by a Taylor series expansion of the sensitivity coefficient:

$$C(p_i, p_j) = C(P_i, P_j) + S_i^{(1)} \Delta\varepsilon_i + S_j^{(1)}\Delta\varepsilon_j + \frac{1}{2!} S_{i,i}^{(2)} \Delta\varepsilon_i^2 + \frac{1}{2!} S_{j,j}^{(2)} \Delta\varepsilon_j^2 + S_{i,j}^{(2)} \Delta\varepsilon_i\Delta\varepsilon_j + h.o.t. \qquad (4)$$

where $C(P_i,P_j)$ is concentration in the base-case simulation, and the higher order term greater than third-order were summarized into h.o.t. The zero-out contribution (ZOC) is defined as the difference between the base-case simulation and the concentration that would occur if the sensitivity parameter did not exist (Cohan et al., 2005). It is derived as follows:

$$ZOC(P_i, P_j) = C(P_i, P_j) - C(p_i = 0, p_j = 0) \approx S_i^{(1)} + S_j^{(1)} - \frac{1}{2} S_{i,i}^{(2)} - \frac{1}{2} S_{j,j}^{(2)} - S_{i,j}^{(2)} \qquad (5)$$

Throughout this study, we investigate the emission impacts based on this ZOC formulation in Eq. (5). The emissions of the $O_3$ precursor species $NO_x$ and non-methane volatile organic compounds (NMVOCs; hereafter simply referred to as VOCs) are used as sensitivity parameters (i and j). For example, the expression of $S_{NO_x}^{(1)}$ means the first-order sensitivity of $O_3$ to $NO_x$ emission.

In addition, HDDM was extended to examine the sensitivity of $O_3$ mixing ratios towards stratospheric $O_3$. A dynamic $O_3$/PV function considering the seasonal, latitudinal and altitude dependencies is constructed at three vertical levels of 58, 76, and 95 hPa fitted as a 5th order polynomial function, and applicable between the range of 50 and 100 hPa (Xing et al., 2016).



The sensitivity to this stratospheric O₃ can be directly calculated, and quantify the effects of STT on tropospheric O₃. This sensitivity is hereafter referred to as O3VORT.

## 4 Results and Discussion of Sensitivity Analysis by HDDM

### 4.1 Sensitivity Regime in April 2010

Sensitivity coefficients towards domain-wide emissions (i.e., emissions across the entire simulation domain) calculated by HDDM are shown in Fig. 1; these values represent monthly-means and in turn are computed from hourly sensitivity coefficients output by the CMAQ model configured with HDDM. Generally, the response of O₃ to NOₓ emissions exhibits positive first-order sensitivities (Fig. 1 (a)) and negative second-order sensitivities (Fig. 1 (c)) because of the concave response of O₃ to NOₓ emissions. Exceptions are found over eastern China to the Korean Peninsula, some parts of Europe, and some cities in the western U.S.A. (e.g., Seattle, San Francisco, and Los Angeles), around the Great Lakes, and the northeastern U.S.A. (e.g., New England region). These regions show negative first-order sensitivity to NOₓ emissions due to the NO titration effect by dense NOₓ emission sources. The values of sensitivity coefficients to VOC emissions (Figs. 1 (b) and (d)) are small compared to those to NOₓ emissions. In addition, the second-order sensitivity coefficients of O₃ to VOCs emissions are also smaller, indicating that the non-linear response of large-scale O₃ distributions to VOC emissions is negligible. A positive second-order cross sensitivity of O₃ to domain-wide NOₓ and VOC emissions (Fig. 1 (e)) demonstrates O₃ will become less responsive to NOₓ emissions with a concurrent reduction of VOCs emissions, and vice versa. While these sensitivities were calculated towards total (i.e., both anthropogenic and biogenic) emissions, the main interest from a policy making perspective is on the sensitivities towards anthropogenic emissions. To estimate these sensitivities, we recalculated sensitivity coefficients of O₃ to isoprene emissions as a proxy for biogenic emissions (Fig. S1). By comparing these sensitivities to isoprene emissions (Fig. S1) to the sensitivities towards all emissions (Fig. 1), it can be concluded that the impacts of biogenic VOCs emissions during April 2010 are small compared to the impacts of NOₓ emissions.

Determining the O₃ sensitivity regime can provide useful information to policy makers designing emission reduction strategies by clarifying the relative importance of precursor emissions. Based on the relationships between the sensitivity coefficients, we determined O₃-sensitivity regimes from threshold values revised from previous studies (Wang et al., 2011; Itahashi et al., 2013) as follows.

$$10 \text{ [ppbv]} < S_{VOCs}^{(1)} \text{ and } S_{NO_x}^{(1)} < S_{VOCs}^{(1)} : \text{ VOC sensitive}$$

$$10 \text{ [ppbv]} < S_{NO_x}^{(1)} \text{ and } S_{VOCs}^{(1)} < S_{NO_x}^{(1)} : \text{ NOx sensitive}$$

Grid cells meeting neither of these two criteria are considered to be in a transition regime. This classification is applied to all hourly HDDM results during April 2010 and then averaged. The O₃-sensitivity regimes obtained through this analysis are shown in Fig. 2. The shading of NOₓ (purple) or VOC (green) sensitive indicates the high frequency of occurrence of sensitivity





to NO$_x$ or VOC Regime. As already suggested by the relative magnitudes of the sensitivity coefficients towards NO$_x$ and VOCs emissions shown in Fig. 1, O$_3$ during April 2010 is in a NO$_x$ sensitive regime over the mid-latitude northern hemisphere with the exception over the locations that had negative first-order sensitivity to NO$_x$ emission and were classified as VOC sensitive. Therefore, controls on NO$_x$ emissions can be an effective way to reduce surface O$_3$ across almost the entire northern

hemisphere but it may cause an increase of O$_3$ mixing ratios over eastern China and some areas in Europe and the U.S.A. Through the analysis of HDDM results for domain-wide emissions, this section provided an overview of O$_3$ sensitivities and the response of O$_3$ to precursor emissions over the northern hemisphere. The following section further investigates the sensitivity of surface O$_3$ over the U.S.A. by defining different emission source regions over the U.S.A. and East Asia.

**4.2 Emission Impacts from U.S.A. and East Asia at Surface Level**

To investigate the emission impacts from the U.S.A. and East Asia, we defined two source regions as shown in Fig. 3. In this study, East Asia includes China, Taiwan, Mongolia, the Korean Peninsula (North and South Korea), and Japan. We conducted additional HDDM simulations using these two source regions and then calculated their sensitivity coefficients which are shown in Figs. S2 and S3 in the supplemental material. Based on these sensitivity coefficients, ZOC of emissions from the

U.S.A. and East Asia are derived according to Eq. (5) and the resulting emission impacts are shown in Fig. 3. The ZOC of emissions from the U.S.A. show more than 10 ppbv over the southeastern U.S.A. and relatively small impacts around 2-8 ppbv in the western U.S.A. In some areas that are characterized by a VOC-sensitive regime in Fig. 2 (e.g., Seattle, San Francisco, Los Angeles, around the Great Lakes, and New England regions), emissions from the U.S.A. have small negative impacts. The U.S.A. emission impacts extend to the Atlantic Ocean with impacts of more than 2 ppbv which are comparable to those found

over the western U.S.A., and then decrease over Africa. The ZOC of emissions from East Asia also shows positive impacts greater than 10 ppbv over China, Taiwan, Japan, and the western Pacific Ocean with the exception of negative impacts over eastern China. These negative impacts indicate that the elimination of emissions can lead to O$_3$ increase, because NO titration works to reduce O$_3$ mixing ratio over these areas that have a high emission density. The analysis of the ZOC from East Asian emissions clearly illustrates the presence of trans-Pacific transport of O$_3$. This transport on a monthly mean basis is estimated

to be more than 2 ppbv over almost the entire Pacific Ocean, and reaches many parts of North America, i.e., almost the entire U.S.A. and Canada and western Mexico.

Detailed analyses of these impacts over the U.S.A. are conducted by focusing on longitudinal differences. In this study, we use four time zones of Pacific, Mountain, Central, and Eastern standard time (abbreviated as PST, MST, CST, and EST, respectively) in the U.S.A. and investigate O$_3$ mixing ratio and ZOC of emission from U.S.A. and East Asia in these

zones. Results for monthly and daily means are shown in Fig. 4 and are also listed in Table 1. Consistent with previous studies (e.g., Simon et al., 2015), O$_3$ mixing ratios have a longitudinal gradient with lower values in the West and higher values in the East (modeled monthly mean concentrations are 35.8, 39.3, 39.1, and 40.6 ppbv over PST, MST, CST, and EST, respectively; see Table 1). The results of the ZOC analysis reveal varying impacts from U.S.A. and East Asia emissions across the four



regions. For the U.S.A on a monthly-mean domain-wide basis, the impact of domestic emissions surpasses that of East Asian emissions. Over the PST region, the monthly averaged impact of domestic emissions is 3.2 ppbv while that of East Asian emissions is 2.8 ppbv, i.e., the impacts from both source regions over the PST zone are comparable. It should be noted that the daily averaged impact of East Asian emissions can exceed that of U.S.A. emissions on some days (e.g., in early April and

during April 27-30), suggesting the significant role of episodic trans-Pacific transport on air quality over the western U.S.A. In contrast to the situation over the PST zone, the impact from domestic emissions always clearly exceeds the impact from East Asian emissions in the MST, CST, and EST zones; this feature strengthens towards the east. For example, the temporal variations of daily averaged $O_3$ mixing ratios and the impacts of domestic emissions are well correlated over the EST zone. The impact of East Asian emissions is small compared to that of U.S.A. emissions over the CST and EST zones, but it is not

negligible. These impacts are 2.1 ppbv on a monthly average basis (ranging between 1.2 ppbv and 3.0 ppbv on a daily basis) over CST and 1.9 ppbv on a monthly average basis (ranging between 1.2 ppbv and 2.8 ppbv on a daily basis) over EST through April 2010.

To illuminate the relationship between surface $O_3$ mixing ratio and impacts from U.S.A. and East Asian emissions, in Figure 5 scatter plots were constructed using model derived estimates at all CASTNET sites and at elevated CASTNET

sites only (refer Fig. 12 of Itahashi et al., 2019). The statistical analysis of correlation coefficient (R) and its significance level by Student's t-test between surface $O_3$ mixing ratio and these impacts by emissions are listed in Table 1. At all CASTNET sites, the relationship between the modeled MD8O3 and the impact of emissions from the U.S.A. shows positive slope with R of 0.63 and p < 0.001; confirming that domestic emissions are generally the cause of high surface $O_3$ mixing ratios. On the other hand, the relationship between modeled MD8O3 and the impact of emissions from East Asia is flat with R of −0.03 and

not significance; suggesting that constant impacts are found in the U.S.A. but do not directly relate to high surface $O_3$ mixing ratios. A noticeable result is that the relationship varies across the regions. Each point in the scatter plots is shaded by time zone, and it can be seen that high $O_3$ mixing ratios over the CST and EST zones (darker black in Fig. 5 (c)) are not linked to the impacts of East Asian emissions (R were 0.06 and −0.03 respectively, and not significant) while moderately higher $O_3$ mixing ratio found over PST and MST (lighter black in Fig. 5(c)) appear to be linked to higher impacts from East Asian

emissions (R were 0.36 and 0.36 respectively, and p < 0.001). These analyses are repeated using data from sites with an elevation higher than 1000 m (see Table S1 in the supplemental material). At this subset of stations, the $O_3$ mixing ratio shows a positive relationship with emissions from both the U.S.A. (R of 0.52 with p < 0.001) and East Asia (R of 0.22 with p < 0.001). This might be partly because most of the elevated CASTNET sites are located in the western U.S.A. (17 of 21 elevated sites are located in the PST or MST zones). Since long-range transport occurs aloft and since changes in pollutant concentrations

influence their ground-level values (e.g., Mathur et al., 2018b) in the next section we specifically investigate the impacts of emissions from different source regions on $O_3$ aloft.





### 4.3 Emission Impacts on O₃ aloft

In this section we focus on the impacts of U.S.A. and East Asian emissions on $O_3$ distributions through the troposphere over the U.S.A. Monthly averaged $O_3$ mixing ratios and ZOC of emissions from the U.S.A. and East Asia at different altitudes in the free troposphere are shown in Fig. 6. Throughout this study, we define the free troposphere to range from 750 to 250 hPa and refer to pressure levels of 750 hPa, 500 hPa, and 250 hPa as the bottom, middle, and top of the free troposphere, respectively. The results of this analysis are also summarized in Table 2. $O_3$ mixing ratios are larger over continents from the surface to 750 hPa (i.e., boundary layer), but are more dispersed over mid to high latitudes at 500 and 250 hPa (Fig. 6). $O_3$ mixing ratios at the surface exhibit a longitudinal gradient with lower values over the western U.S.A. and higher values over the eastern U.S.A., and the same gradient is seen at 750 hPa. However, there are no longitudinal gradients at 500 hPa with 54 ppbv over the entire U.S.A., and a reversed longitudinal gradient with western highs and eastern lows is found at 250 hPa (Table 1).

Once $O_3$ is lofted to free troposphere, its sinks are not effective and consequently it can be transported further. For ZOC of U.S.A. emissions, the largest contribution is found over the southeast U.S.A. at 750 hPa but the impacts of U.S.A. emissions stretch far across the Atlantic to Europe, North Africa, Eurasia, and even to Japan with values above 2 ppbv. Areas where the impact of U.S.A. emissions exceeds 2 ppbv are shown over the entire northern hemisphere at 500 and 250 hPa (Fig. 6). It should also be noted that the impacts of U.S.A. emissions on U.S.A. remained constant or declined with increasing altitude. In particular, constant impacts from U.S.A. emissions with increasing altitude are found over the PST zone, whereas decreasing impacts are found over the MST, CST, and EST zones. From the middle to the top of the free troposphere, the impacts of U.S.A. emissions on U.S.A. are around 2-3 ppbv (Table 2). For ZOC of East Asian emissions, extended impacts on U.S.A. when increasing altitude are shown (Fig. 6). At 750 hPa, the impacts are found over the entire Pacific Ocean with more than 10 ppbv around Hawaii and contribution as high as 4-8 ppbv over the entire U.S.A. At 500 hPa, its impacts are smaller over the Pacific Ocean with less than 8 ppbv; however, the impacts are above 6 ppbv almost across the entire U.S.A., surpassing the impacts found at 750 hPa. At 250 hPa, the impacts are slightly decreased beyond the U.S.A., but stretched across a broader range to Europe and western Russia (Fig. 6). It is shown that the impacts of East Asian emissions are around 5 ppbv or more over the entire free troposphere over U.S.A. (Table 2). From the middle to the top of the free troposphere, the impacts of emissions from East Asia are twice or more those of U.S.A. emissions over the eastern and western U.S.A., respectively.

In characterizing the dominant sources of $O_3$ aloft, the role of stratospheric air masses also needs to be considered. In our part 1 paper, we developed an air mass characterization technique, but it was limited to estimate the air mass burden on column $O_3$. In this part 2 paper, to unify the methodology investigating sensitivities to model parameters, the sensitivity towards $O_3$ specification near the tropopause based on a potential vorticity scaling, hereafter referred to as O3VORT, is directly calculated. The results of the $O_3$ sensitivity towards O3VORT are shown in Fig. 7 at the surface, 750 hPa, 500 hPa, and 250 hPa with different color scales. Not surprisingly, the sensitivity of $O_3$ to O3VORT shows increasing values with increasing altitude. At the surface level and on a monthly averaged time scale, the impact of STT is less than 1 ppbv except over the Tibet





plateau because of its elevation. In other regions, smaller impacts of STT are noted over the western U.S.A. and north Africa; the former due to the high elevation of Rocky mountain whereas the latter is likely related to active convection. Impacts of STT exceeding 1 ppbv are found over mid-latitudes areas at 750 hP, and stronger impacts exceeding 5 ppbv are found at 500 hPa. At 250 hPa, the impacts of STT are shifted towards high-latitudes and exceed 25 ppbv, reflective of the lower tropopause height at higher latitudes (Fig. 7). Over the U.S.A., the monthly averaged impacts of STT are below 1 ppbv at the surface and 750 hPa and increase from around 2 ppbv at 500 hPa to more than 10 ppbv at 250 hPa (Table 1). At 250 hPa, the impacts of STT range from more than 20 ppbv in the west and a low of around 10 ppbv in the east.; therefore, these differences partly account for the longitudinal gradient of the $O_3$ mixing ratio modeled at the top of free troposphere.

To illustrate altitude dependencies of the impacts of U.S.A. and East Asian emission and STT, vertical cross-sections ("curtain plots") of these impacts at six ozonesonde sites across the U.S.A. are examined in Fig. 8 (refer Figs. 4 and 10 of Itahashi et al., 2019). In these curtain plots, the pressure levels of 750, 500, and 250 hPa are marked to indicate the representative altitude of the bottom, middle, and top of the free troposphere. The comparison of the ZOC from U.S.A. and East Asian emissions clearly shows the differences of their vertical structures. Over these ozonesonde sites except Hilo (Fig. 8 (a)), the emission impacts from the U.S.A. greater than 10 ppbv are mostly confined below 750 hPa (within the boundary layer) and occasionally extend into the free troposphere. In contrast, the emission impacts from East Asia can predominantly be found in the free troposphere and sometimes extend into the boundary layer (below 750 hPa) and/or the upper model layers (above 250 hPa). These patterns further confirm that pollution lofted to the free-troposphere over Asia can undergo efficient transport across the Pacific and entrain to the lower troposphere and boundary layer over the U.S. The sensitivity towards O3VORT is the dominant factor over the upper model layers (above 250 hPa) and downward into the upper part of the free troposphere, but most of its episodic impact does not reach to the middle of the free troposphere (500 hPa) or below. The strong STT events seen in these cross-sections, i.e., as the events in early and late April at Trinidad Head (Fig. 8 (b)), early April at Boulder (Fig. 8 (c)), late April at Huntsville (Fig. 8 (d)), and middle April at Wallops Island (Fig. 8 (e)) and Rhode Island (Fig. 8 (f)) are generally consistent with the results inferred from the airmass classification technique presented in the part 1 paper. It should be however noted that a more robust quantification of the fraction of ground level $O_3$ that originated in the stratosphere and its seasonal and spatial distributions would require conduct of longer-term sensitivity simulations than those examined here.

## 4.4 Perspective on the Changes in Trans-Pacific Transport

As has been shown in previous studies and affirmed in the current work, that trans-Pacific transport can impact air quality in the U.S.A. April 2010 was used as the target period for our analysis because El Niño conditions during that time period favored trans-Pacific transport. In this section, we estimate the variation of trans-Pacific transport caused by recent emission changes. According to the NOAA Climate Prediction Center (CPC), strong and long-lasting El Niño conditions occurred from late 2014 to the middle of 2016 (NOAA, 2018). Observed average MD8O3 over the U.S.A. was 46.9 ppbv in



April 2015, a decline from its April 2010 values of 52.2 ppbv, and the number of sites exceeding the NAAQS declined from 39 sites in April 2010 to 7 sites in April 2015. From 2010 to 2015, annual $NO_x$ (VOCs) emissions in the U.S.A. decreased from 13.4 (13.6) Tg to 10.6 (12.9) Tg (see, Fig. S1 of supplemental material of Itahashi et al., 2019). These emission reductions likely contributed to the decline of the observed $O_3$ mixing ratios relative to the 2010 values. How about the trans-Pacific

transport? Anthropogenic emissions in China grew during the 2000s (Itahashi et al., 2014), and reached the highest levels in the world in 2010; however, substantial reductions have been measured by satellites since then (Irie et al., 2016; Krotkov et al., 2016; van der A et al., 2017; Itahashi et al., 2018). In addition, bottom up emission inventories indicate that Chinese $NO_x$ emissions were reduced as consequence of clean air actions (Zheng et al., 2018). In particular, Zheng et al. (2018) report that annual $NO_x$ emissions were reduced from 26.5 Tg in 2010 to 23.7 Tg in 2015 while annual VOC emissions increased from

25.9 Tg in 2010 to 28.5 Tg in 2015. While $NO_x$ emissions have been regulated and subsequently declined after reaching a peak of 29.2 Tg in 2012, the situation is more complex for VOCs emissions which show decreases from the residential and transportation sectors but increases from the industrial sector and solvent use. Applying the percentage changes in Chinese emissions from 2010 to 2015 to the HDDM sensitivities for East Asian emissions (assuming that changes in East Asian emissions are dominated by changes in China), we estimated their impacts on tropospheric $O_3$ mixing ratios.

15       The changes in $O_3$ mixing ratio caused by emission changes between 2010 and 2015 over the U.S.A. and East Asia can be investigated via Eq. (4). Based on the emission changes noted above, the resulting values of $\varepsilon_i$ and $\varepsilon_j$ in Eq. (4) are -20.9% and -5.1% for $NO_x$ and VOC emissions from the U.S.A., and -10.6% and 10.0% for $NO_x$ and VOC emissions from East Asia, respectively. The estimated spatial changes in $O_3$ mixing ratios at the surface and aloft are shown in Fig. 9, and estimates for monthly and daily means over four time zones in the U.S.A. are shown in Fig. 10 in a similar manner to Fig. 4. The U.S.A.

emission reductions between 2010 and 2015 resulted in generally reducing surface $O_3$ mixing ratios with changes of ~~around~~ at least −0.5 ppbv across the entire U.S.A. and up to −5.0 ppbv over the southeast U.S.A. Exceptions are found over Seattle, San Francisco, Los Angeles, around the Great Lakes, and in New England regions that were characterized as VOC sensitive in Section 3.2. These changes are expected because reductions in $NO_x$ emission were greater than those in VOC emissions across the U.S. It is also shown that the U.S.A. emission reductions cause a reduction of $O_3$ mixing ratio over the free troposphere.

On the time zone averaged basis, the changes in monthly-mean $O_3$ mixing ratio are −0.5, −1.1, −1.8, and −1.5 ppbv over PST, MST, CST, and EST, respectively (Fig. 10). The maximum reduction are found over CST because EST contains the complex sensitivity over New England regions. In contrast, the changes in East Asian emissions between 2010 and 2015 do not cause a noticeable reduction in surface $O_3$ mixing ratios over the U.S.A. while they lead to $O_3$ mixing ratio increases of more than 1 ppbv over eastern China, the Korean Peninsula, and some parts of Japan on monthly average (Fig. 9). These increases are

expected both because these areas were shown to be VOC-sensitive in Section 3.2 and because of the increase in VOC emissions. On time zone averaged basis, changes in East Asia emissions between 2010 and 2015 are estimated to change monthly mean $O_3$ mixing ratio across the U.S. by about −0.1 to −0.3 ppbv. The corresponding changes in daily average surface-level $O_3$ mixing ratio were also less than −0.5 ppbv (Fig. 10). A slight reduction in monthly mean $O_3$ mixing ratios of around −0.5 ppbv was estimated across large parts of the Northern Hemisphere free troposphere, indicating that the reductions in East



Asian emissions that occurred between 2010 and 2015 can partly contribute to a weakening of trans-Pacific $O_3$ transport over the free troposphere. However, the reductions in Asian emissions during 2010 and 2015 did not appear to alter monthly means surface levels $O_3$ mixing ratio across the U.S.A.

**5 Conclusions**

In this study, the regional chemical transport model extended for hemispheric applications, H-CMAQ, is applied to investigate trans-Pacific transport during April 2010. A previous part 1 manuscript demonstrated that STT can cause impacts on tropospheric $O_3$, but did not relate to the enhancement of surface $O_3$ mixing ratios. Therefore, in this part 2 manuscript, emission impacts are investigated based on the sensitivity analysis through HDDM. The sensitivities to domain-wide emissions

indicate $NO_x$-sensitive conditions during April 2010 for tropospheric $O_3$ across most of the Northern Hemisphere except over eastern China and a few urban areas over the U.S. and Europe. Contributions of emissions from source regions covering the U.S.A. and East Asia were examined through propagation of emission sensitivities in HCMAQ. Analysis of estimated zero-out contributions from the computed sensitivities demonstrate comparable impacts of U.S.A. and East Asian emissions on surface level $O_3$ over the western U.S.A. during April 2010 whereas contributions from U.S.A. emissions dominate $O_3$

distributions over the eastern U.S.A. The analyses also reveal the significant impacts of East Asian emissions on free tropospheric $O_3$ over the U.S.A. which surpass the estimated impacts of U.S.A. emissions, further confirming the long-range pollution transport conceptual view wherein pollution from source regions is convectively lofted to the free troposphere and efficiently transported intercontinentally. Finally, the effects of recent emission changes on the trans-Pacific transport of $O_3$ are estimated. Under the assumed similar meteorological condition on 2010 and 2015, it can be concluded that trans-Pacific

transport resulting from emission changes did not lead to significant changes in $O_3$ mixing ratio over U.S.A. at the surface level even on a daily mean basis in April. The year 2015 was selected because of El Niño conditions favorable to trans-Pacific transport, however, the impacts of changes in year specific meteorological conditions are not investigated here. The possible impacts of changing climate on trans-Pacific transport (e.g., Glotfelty et al., 2014) should however be further examined. Long-term trend analysis taking into accounts both emission and meteorological changes (e.g., Mathur et al., 2018a) will be studied

in future work to further understand variability in trans-Pacific transport patterns and contributions. While the one-month simulation period and analysis of a representative spring-time month helped characterize aspects of trans-Pacific transport, longer term simulations need to be conducted to further quantify the seasonal source region contributions to trans-Pacific transport. The results presented here are based on monthly or daily mean ozone during April and are not expected to be consistent with other metrics (e.g., MD8O3) or times of the year when transport is less favorable and local ozone production

is more favorable. The longer-term calculations will also help better quantify the STT contributions to surface-level $O_3$ which appear to be lower in the current analysis relative to previous studies (e.g., Lelieveld and Dentener, 2000; Lin et al., 2015; Mathur et al., 2017).



**Code availability**

Source code for version 5.2 of the CMAQ model can be downloaded from https://github.com/USEPA/CMAQ/tree/5.2. For further information, please visit the US Environmental Protection Agency website for the CMAQ system: https://www.epa.gov/cmaq.

**Data availability**

The code of decoupled direct method on the version 5.2 of the CMAQ model can be downloaded from https://github.com/USEPA/CMAQ/tree/5.2. For further information, please visit the US Environmental Protection Agency website for the CMAQ system: https://www.epa.gov/cmaq.

**Competing interests**

The authors declare that they have no conflict of interest.

**Disclaimer**

15  The views expressed in this paper are those of the authors and do not necessarily reflects the views or policies of the U.S. Environmental Protection Agency.

**Author contributions**

Syuichi Itahashi performed the analysis of observation and model simulation and prepared the manuscript with contributions

20  from all co-authors. Rohit Mathur and Christian Hogrefe contributed to establish the hemispheric modeling application for this

study and prepared the emission dataset and initial condition from previous long-term simulation results. Sergey L. Napelenok

contributed to the discussion of sensitivity analysis based the higher-order decoupled direct method. Yang Zhang contributed

to the literature review of trans-Pacific transport and refined this research through simulation designs, and results interpretation.



## Acknowledgement

Yang Zhang acknowledges support from the 2017-2018 NC State Internationalization Seed Grant and the 2019-2020 NC State Kelly Memorial Fund for US-Japan Scientific Cooperation.

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





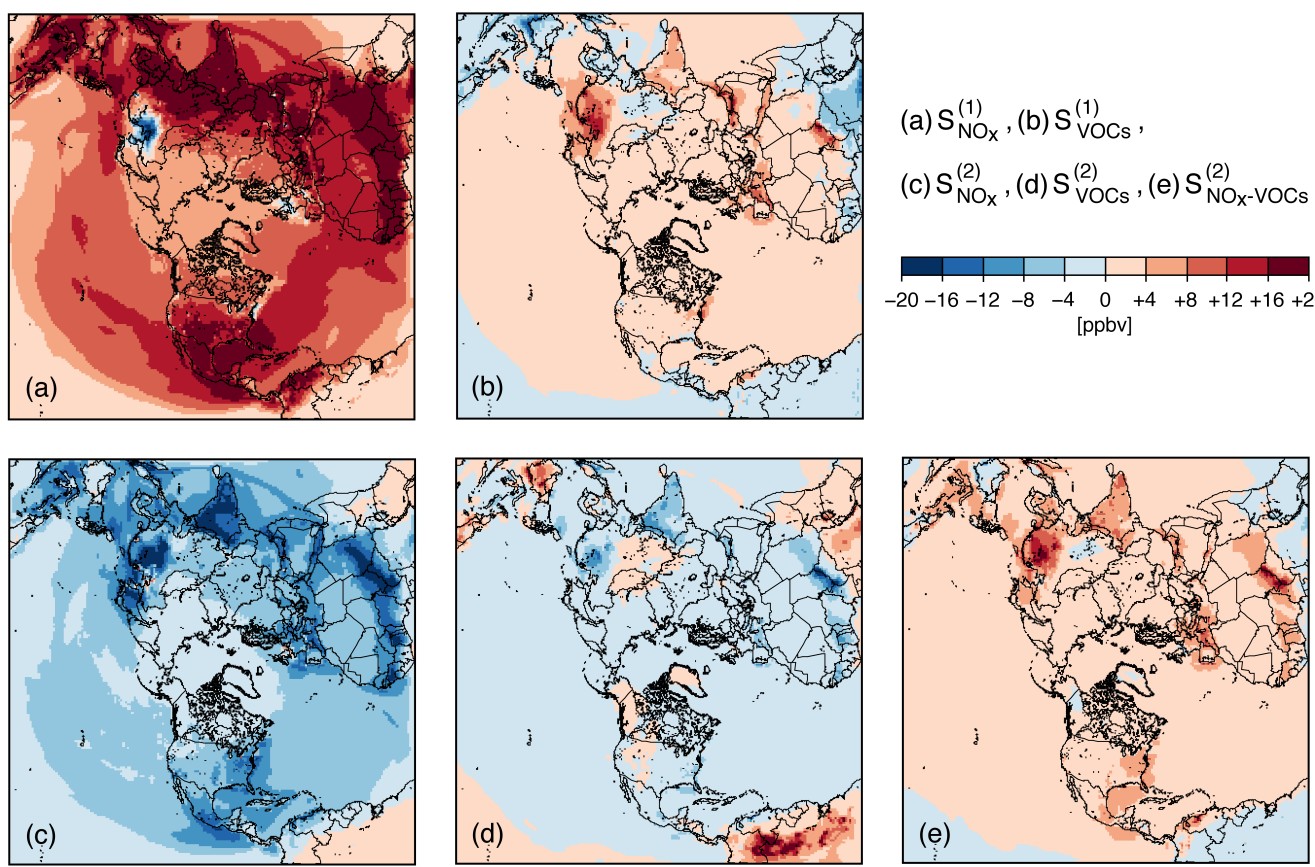

(a) $S^{(1)}_{NO_x}$ , (b) $S^{(1)}_{VOCs}$ ,

(c) $S^{(2)}_{NO_x}$ , (d) $S^{(2)}_{VOCs}$ , (e) $S^{(2)}_{NO_x\text{-}VOCs}$

**Figure 1.** Spatial distribution of the sensitivity coefficients of $O_3$ to domain-wide emissions. (a) first-order sensitivity to $NO_x$ emissions, (b) first-order sensitivity to VOCs emissions, (c) same as (a) but as second-order, (d) same as (b) but as second-order, and (e) second-order sensitivity to $NO_x$ and VOCs emissions during April 2010.





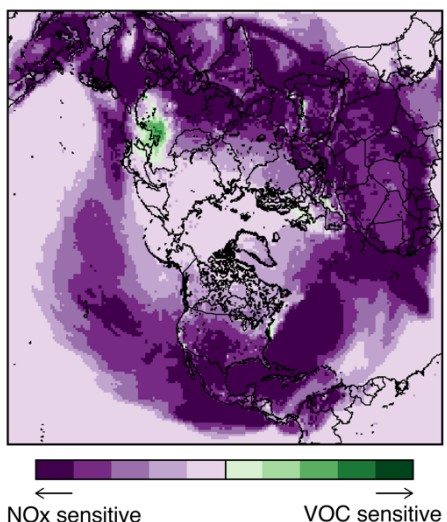

**Figure 2. Spatial distributions of the ozone-sensitive regime during April 2010.**


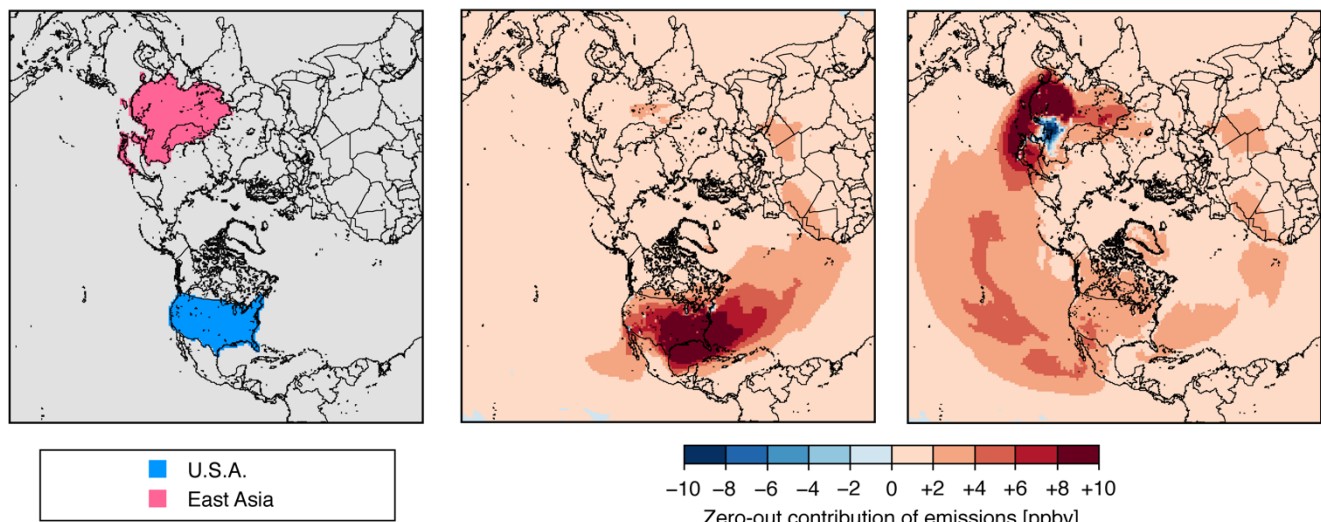

**Figure 3. Source regions of U.S.A. and East Asia (left), and zero-out contribution of emissions from U.S.A (center) and East Asia (right) during April 2010. East Asia are defined as China, Taiwan, Mongolia, Korean Peninsula, and Japan.**





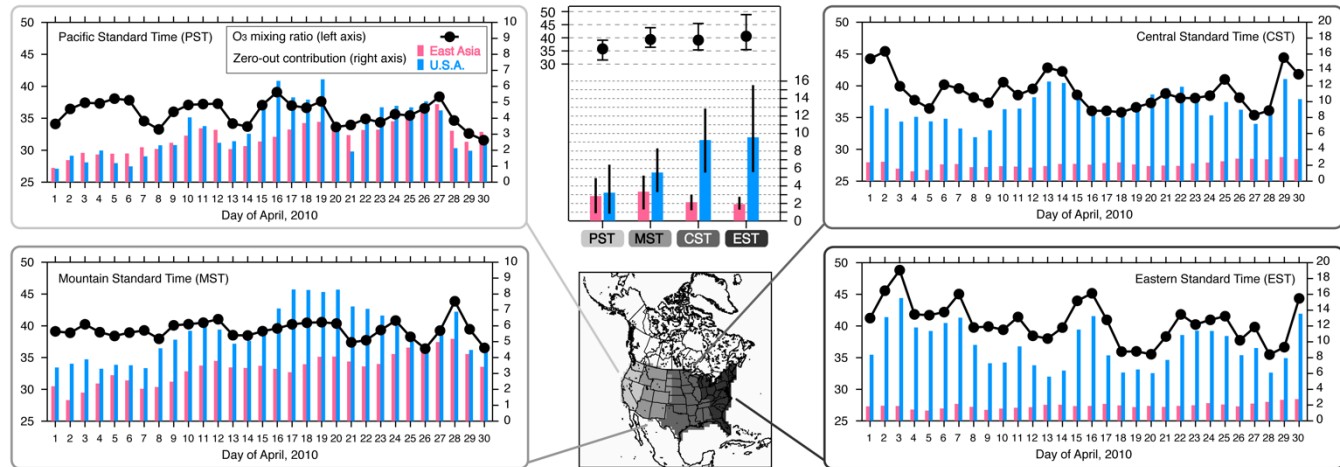

**Figure 4.** Daily and monthly averaged O₃ mixing ratio (left axis; black circles and thick lines) and zero-out contribution from U.S.A. and East Asia (right axis; light blue and light red bars respectively) summarized over four time zones of Pacific, Mountain, Central, and Eastern Standard Time (PST, MST, CST, and EST) in U.S.A. The units of left and right-axis is ppbv. On monthly averaged (center panel), whiskers indicates daily minimum and maximum. Note that the axis of zero-out contributions is different in left (PST and MST) and right panels (CST and EST).



**Figure 5.** Relationship between modeled MD8O3 at the surface and zero-out contribution of emissions from (a, b) U.S.A. and (c, d) East Asia. The points are shaded by four time zones in U.S.A. (a, c) All CASTNET sites, and (b, d) elevated CASTNET sites defined as having an elevation greater than 1000 m (see also Table S1).





**Figure 6. Monthly averaged O₃ concentration (first row) and zero-out contribution from U.S.A. (second row) and East Asia (third row) at bottom of free troposphere (750 hPa; left column), middle of free troposphere (500 hPa; center column), and top of free troposphere (250 hPa; right column).**





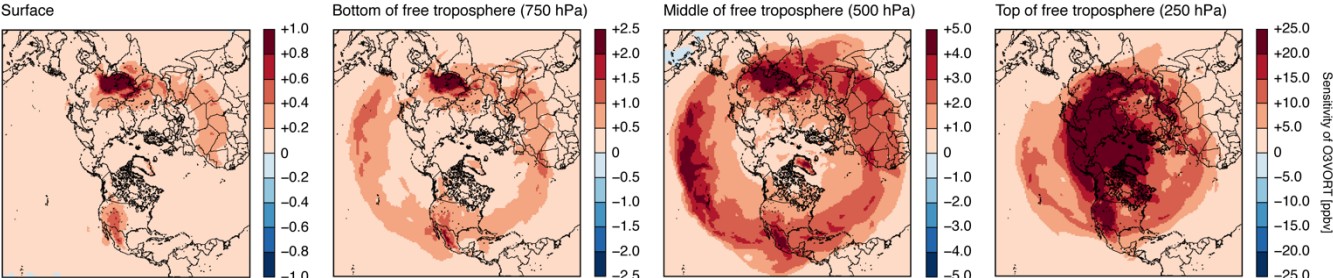

**Figure 7. Monthly averaged sensitivity of O3VORT at surface, bottom of free troposphere (750 hPa), middle of free troposphere (500 hPa), and top of free troposphere (250 hPa) from left to right.**



**Figure 8.** Curtain plots of (left) ZOC of emissions from U.S.A., (center) ZOC of emissions from East Asia, and (right) sensitivity of O3VORT at U.S. ozonesonde sites of (a) Hilo (HI), (b) Trinidad Head (CA), and (c) Boulder (CO) during April 2010. Yellow stars indicate the time of available ozonesonde measurements. Thick lines from bottom to top indicate 750, 500, and 250 hPa as a representative bottom, middle, and top of free troposphere.





**Figure 8. Continued, but at (d) Huntsville (AL), (e) Wallops Island (VA), and (f) Rhode Island (RI).**



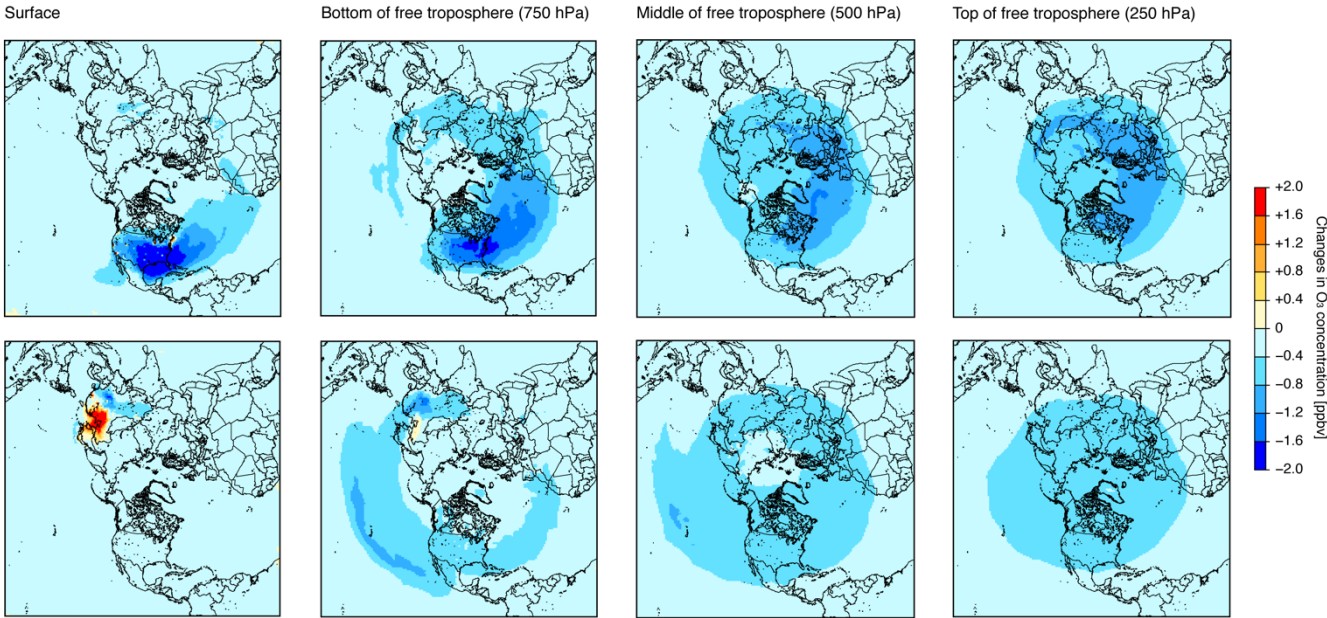

**Figure 9. Changes in O₃ concentration caused by emission changes from (top) U.S.A. and (bottom) East Asia at surface, bottom of free troposphere (750 hPa), middle of free troposphere (500 hPa), and top of free troposphere (250 hPa) from left to right.**



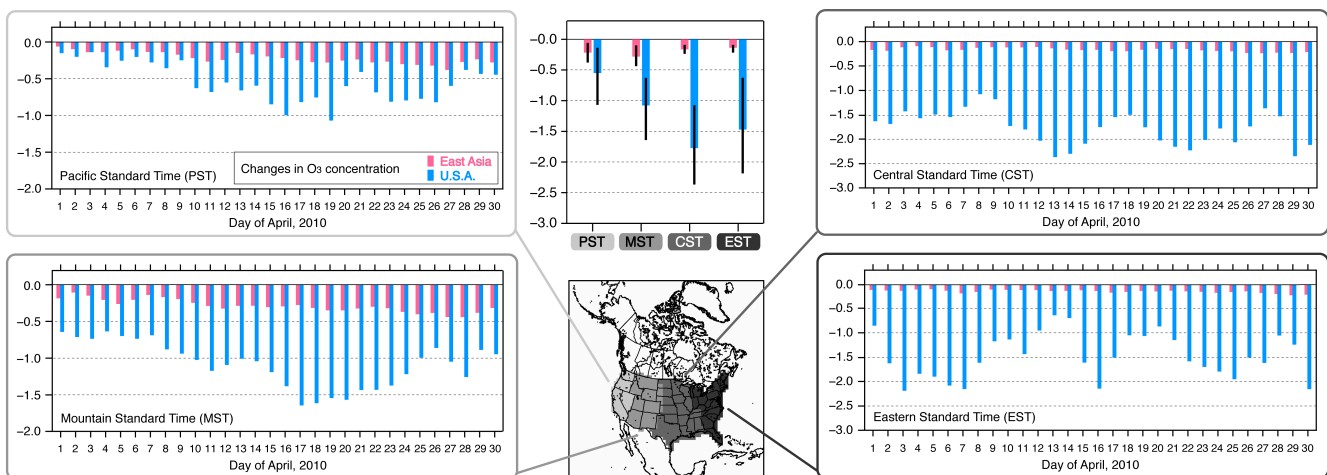

**Figure 10. Daily and monthly averaged changes in O₃ mixing ratio caused by emission changes from U.S.A. (light blue bars) and East Asia (light red bars) summarized over four time zones of Pacific, Mountain, Central, and Eastern Standard Time (PST, MST, CST, and EST) in U.S.A. The units is ppbv. On monthly averaged (center panel), whiskers indicates daily minimum and maximum. Note that the axis is different in left (PST and MST) and right panels (CST and EST).**





**Table 1. Summary of correlation between modeled MD8O3 and zero-out contribution of emissions from U.S.A. and East Asia.**

| | N | Emission impacts from U.S.A. | Emission impacts from East Asia |
|---|---|---|---|
| All CASTNET sites | 2286 | 0.63*** | −0.03 |
| −Pacific Standard Time (PST) | 238 | 0.52*** | 0.38*** |
| −Mountain Standard Time (MST) | 359 | 0.65*** | 0.36*** |
| −Central Standard Time (CST) | 489 | 0.55*** | 0.06 |
| −Eastern Standard Time (EST) | 1200 | 0.64*** | −0.02 |
| Elevated CASTNET sites | 587 | 0.52*** | 0.22*** |

Note: Significance levels by Students' t-test for correlation coefficients between observations and simulations are remarked as $*p < 0.05$, $**p < 0.01$, and $***p < 0.001$, and lack of a mark indicates no significance.



**Table 2. Summary of O₃ concentration and zero-out contribution of emissions from U.S.A. and East Asia, and sensitivity of O3VORT over four time zones in U.S.A. during April 2010.**

|  | O$_3$ concentration | Emission impacts from U.S.A. | Emission impacts from East Asia | Impacts by stratospheric intrusion |
|---|---|---|---|---|
| Pacific Standard Time (PST) |  |  |  |  |
| −Surface | 35.8 | 3.2 | 2.8 | 0.2 |
| −Bottom of free troposphere | 47.3 | 2.7 | 6.1 | 0.7 |
| −Middle of free troposphere | 54.0 | 2.4 | 7.3 | 2.0 |
| −Top of free troposphere | 108.3 | 3.0 | 6.5 | 22.3 |
| Mountain Standard Time (MST) |  |  |  |  |
| −Surface | 39.3 | 5.5 | 3.3 | 0.4 |
| −Bottom of free troposphere | 50.3 | 4.8 | 5.7 | 0.9 |
| −Middle of free troposphere | 54.8 | 2.7 | 7.2 | 2.2 |
| −Top of free troposphere | 119.7 | 3.3 | 6.4 | 28.3 |
| Central Standard Time (CST) |  |  |  |  |
| −Surface | 39.1 | 9.2 | 2.1 | 0.2 |
| −Bottom of free troposphere | 50.9 | 6.6 | 4.9 | 0.6 |
| −Middle of free troposphere | 53.9 | 3.3 | 6.6 | 2.0 |
| −Top of free troposphere | 79.9 | 2.9 | 6.1 | 12.9 |
| Eastern Standard Time (EST) |  |  |  |  |
| −Surface | 40.6 | 9.6 | 1.9 | 0.1 |
| −Bottom of free troposphere | 52.0 | 7.9 | 5.0 | 0.6 |
| −Middle of free troposphere | 53.2 | 3.9 | 6.2 | 2.0 |
| −Top of free troposphere | 78.9 | 3.4 | 6.0 | 12.8 |

Note: All units are ppbv.