# Peer review of "Modeling Trans-Pacific Transport and Stratospheric Intrusion of Tropospheric Ozone using Hemispheric CMAQ during April 2010: Part 2. Examination of Emission Impacts based on the Higher-order Decoupled Direct Method"

_Atmospheric Chemistry and Physics, 2019_

## Referee Comment (RC1) · Anonymous Referee #1 · 1 Oct 2019

This paper presents high-order sensitivity analysis modeling of the impacts of emissions and stratosphere to troposphere transport on ozone. Modeling is done with a relatively recent hemispheric version of a widely-used photochemical model. While the methods are not fully novel, they pull together two relatively advanced techniques: HDDM and hemispheric modeling. Explanations are for the most part clear. I recommend publication after addressing the comments below.

[Figure]

Major comments: 1. It is difficult to reconcile the concentration and zero-out contribution estimates. In Figures 6 and 7 and Table 2, concentrations are much larger than the sums of the ZOCs. For example, over most mid- and high-latitude locations in the NH top of the free troposphere (Fig 6, top, c), ozone mixing ratios are over 75 ppb, but the zero-out contributions of East Asian emissions, USA emissions, and stratospheric ozone add to less than half that much. What's the source for the rest of the O3? Sure, there will be influences from other regions, but I would have expected these to be the largest contributions and for cross-sensitivity interactions to be small. More exploration and discussion are needed on this. 2. The results around the perimeter (i.e., tropics) in Figure 1 are peculiar, showing negative first-order sensitivity to VOC, positive second-order sensitivities, negative cross-sensitivity. All of these are opposite in sign to what the chemistry would typically suggest. Further investigation is needed to explore the role of boundary conditions or other factors in driving this, or if there is an error in the modeling.

Minor comments on text: p. 1, Line 32: The 250 to 50 hPa layer is actually in the stratosphere, so not "stratospheric intrusions" p. 4, line 4: typo p. 6, line 1: It is not clear to me how the sensitivity to stratospheric O3 is being calculated. p. 6, line 10: It should be noted that because the coarse grid resolution smears out NOx, you may be missing locations where O3 is actually VOC-limited, such as urban cores with intense NOx emissions at subgrid scales. p. 6, lines 21-22: It would be more appropriate to say: ". . . it can be concluded that ozone is more sensitive to NOx emissions than to biogenic VOCs emissions during April 2010." Also, at some point you should note that not all NOx is anthropogenic (e.g., lightning, soils). p. 10, line 2: Have you identified evidence of "active convection" in the meteorological model, or is this mere speculation? p. 12, line 29: It is difficult to follow where results are being presented on a MD8O3 or 24-hour average basis. Those sensitivities can be quite different.

Specific comments on figures: Fig 1: In the caption, clarify if these are for 8-hour maximum or daily-average results. Fig 9 and 10: Caption needs to say what the emissions

changes were.

---

## Referee Comment (RC2) · Anonymous Referee #2 · 2 Oct 2019

Itahashi et al. (2019) investigated the impacts of emissions from East Asia and US on surface O3 over Northern Hemisphere using HDDM in H-CMAQ. They found comparable impacts by the emissions from East Asia and US on surface ozone over western US with US domestic emissions having larger impacts on surface ozone over eastern US while emissions from East Asia have much larger impacts on free troposphere through trans-pacific transport. But they also found the impacts of recent emission changes in East Asia on US O3 levels were small. The manuscript is in general well written but

there are several concerns about the methodology used in the manuscript.

ZOC of emissions

The estimation on the emission impacts is based on only one H-CMAQ simulation, is that correct? If so, the ZOC estimates are not really accurate as if we reduce all the targeted emissions, meteo/dynamics will also change, which could have feedbacks on the estimations of the emission impacts. The atmospheric conditions will be different under normal and ZOC scenarios. Any explanations on this?

Also, ZOC is not a really realistic scenario and the impacts of ZOC are probably over-estimated in this work. So your conclusion about the impacts of the emissions from East Asia may not be that robust. As in your last part of the work, the impacts of recent changes in the emissions from East Asia were found to be small to insignificant. Unless all the emissions from East Asia were removed, significant impacts would exist over western US at surface and entire US in free troposphere? In other words, To what degree of the changes in the emissions of East Asia should be achieved to show noticeable impacts on US?

For ZOC scenario, does it simply remove all the NOx and NMVOCs emissions? How to distinguish between anthropogenic and natural emissions for NMVOCs? Are the impacts of ZOC the total impacts from the removal of both NOx and NMVOCs? Can we have individual impacts from the removal of NOx only and removal of NMVOCs only? Or maybe we can infer from the sensitivity to these emission? This information could provide more guidance on future emission mitigations.

Specific comments:

Page 1, Abstract, line 24, "with a magnitude of about 3 ppbv impacts on a monthly mean . . ." 3 ppbv O3?

Page 4, line 4, "O3 mixing rations and an those of inert tracer. . .", grammatical error

Page 5, line 5-6, equation (3), should Si,j(2) have the unit of square of the concentration?

Page 6, and Figure 1, any physical meaning on second-order sensitivity? line 9-10, can you explain more on "concave response"?

Page 6, line 27-28, what do you mean by "Svocs and Snox"? it is not equal to Svocs + Snox, right? Maybe you mean Svocs-nox ? Please clarify.

Page 7, line 17-18, regions in VOC-sensitive regime are not clearly shown in Fig 2. You may want to change the color scale to improve the quality.

Page 21, Figure 3, are these impacts based on zero-out of both NOx and NMVOCs at the same time? See my general comments above.

---

## Author Comment (AC1) · 17 Dec 2019

Response to Referee Comment 1 by Anonymous Referee #1

This paper presents high-order sensitivity analysis modeling of the impacts of emissions and stratosphere to troposphere transport on ozone. Modeling is done with a relatively recent hemispheric version of a widely-used photochemical model. While the methods are not fully novel, they pull together two relatively advanced techniques: HDDM and hemispheric modeling. Explanations are for the most part clear. I recommend publication after addressing the comments below.

**Reply:**

**We thank the reviewer for the overall positive assessment of the manuscript and providing helpful and constructive comments. We have revised our manuscript according to the reviewer's comments and suggestions. We believe that these revisions address all points raised by the reviewer. Our point-by-point responses are provided below, and revisions are indicated in blue in the revised manuscript.**

Major comments:

1. It is difficult to reconcile the concentration and zero-out contribution estimates. In Figures 6 and 7 and Table 2, concentrations are much larger than the sums of the ZOCs. For example, over most mid- and high-latitude locations in the NH top of the free troposphere (Fig 6, top, c), ozone mixing ratios are over 75 ppb, but the zero-out contributions of East Asian emissions, USA emissions, and stratospheric ozone add to less than half that much. What's the source for the rest of the O3? Sure, there will be influences from other regions, but I would have expected these to be the largest contributions and for cross-sensitivity interactions to be small. More exploration and discussion are needed on this.

**Reply:**

**We agree with the reviewer that we should have provided additional discussion on the differences between the concentrations and the zero-out contributions for USA emissions, East Asia emissions, and stratospheric ozone in Table 2 and Figures 6-7. To examine the impacts from other regions except U.S.A. and East Asia, we have added the figure illustrating the ZOC of domain-wide emissions at surface and different altitude in Figure S5. In addition to the emissions contributions from other regions, a large part of the difference is due to the effects of initial conditions of both $O_3$ and other species, particularly reservoir species. To illustrate these effects, we have conducted an additional DDM simulation to investigate the sensitivity of $O_3$ to $O_3$ initial and boundary conditions. We have added new supplemental Figures S6-S9 to present these results as well as the results of the domain-wide emissions zero-out contributions over the regions and layers shown in Table 2. These figures show that the domain-wide emission zero-out contributions (Figure**

S5) are larger than the U.S.A. and East Asia zero-out contributions in (Figure 6 and Table 2) (e.g. roughly 30 ppb at the top of the free atmosphere over the Pacific Time Zone (PST) region for April shown in Figure S5 vs. 3.0 + 6.5 ppb = 9.5 ppb for the U.S.A. and East Asia zero-out contributions in Table 2), pointing to the impact of emissions from these other regions on simulated ozone concentrations. Furthermore, while the contribution of $O_3$ initial and boundary conditions decreases over time as the domain-wide emissions zero-out contribution increases, it remains substantial throughout the analysis period. The following paragraph has been added to Section 4.3.

"Note that $O_3$ concentration fields and the sum of sensitivities do not generally equal each other because of nonlinearities in $O_3$ formation. Moreover, the zero-out contributions for U.S.A. and East Asia emissions represent only a portion of the total emissions burden, and the emissions sensitivity calculations can also be affected by initial and boundary conditions. To investigate this further, the temporal evolution of $O_3$ concentrations and sensitivities towards O3VORT, O3IC, O3BC and domain-wide emissions ZOC are presented in Figs. S6-9. The Figures show time series of these contributions averaged over the PST, MST, CST, and EST areas in the U.S.A. at the surface, 750 hPa, 500 hPa, and 250 hPa, corresponding to the results presented in Table 2. These figures show that the domain-wide emission zero-out contributions (Figure S5) are larger than those of zero-out contributions from U.S.A. and East Asia (Figure 6 and Table 2), pointing to the impact of emissions from other regions on simulated ozone concentrations. As expected, the impact of O3BC is small over the U.S.A due to the distance from the equatorial boundaries. At the beginning of the simulation, $O_3$ concentrations are dominated by initial conditions as shown by the close agreement between the $O_3$ concentration and O3IC curves during the first half of March. The sensitivity towards O3IC is declining throughout the simulation while O3VORT and ZOC are increasing and begin to dominate the $O_3$ variation by April. However, even after the one-month spin-up period, O3IC are still present over all time zones and all altitudes. In this study, we initiated the H-CMAQ simulation from the prior model simulation for 2010 (Hogrefe et al., 2018); however, this result suggest that spin-up periods longer than one month may be necessary to fully capture the effects of emissions and O3VORT contributions through calculating HDDM sensitivities over a hemispheric-scale modeling domain. Finally, Figures S6–S9 still show differences between simulated concentrations and the sum of O3VORT, O3IC, O3BC, and ZOC. Aside from the non-linearities and interactions mentioned above, this likely is also caused by contributions of initial conditions of species other than $O_3$ (e.g., PAN or $N_2O_5$) to the simulated $O_3$ levels."

2. The results around the perimeter (i.e., tropics) in Figure 1 are peculiar, showing negative first-order

sensitivity to VOC, positive second-order sensitivities, negative cross-sensitivity. All of these are opposite in sign to what the chemistry would typically suggest. Further investigation is needed to explore the role of boundary conditions or other factors in driving this, or if there is an error in the modeling.

**Reply:**

**We appreciate this insightful comment. The negative first-order sensitivity to VOC and the positive second-order sensitivity to VOC and positive cross sensitivity found in Figure 1 likely were affected by the boundary conditions. To clearly state this perimeter sensitivity to VOC, we have added the following sentence in Section 4.1.**

**"It should be also noted that the positive first-order and negative second-order sensitivities to VOC found near the lateral boundary with ring-shape in the modeling domain could be the perimeter sensitivity. In this H-CMAQ modeling system, the boundary conditions are taken from the clean tropospheric background values with updates to the physical and chemical sinks for organic nitrate species (Mathur et al., 2017). For these boundary conditions, the NO concentration was set to zero, the NO$_2$ concentration was set to 10$^{-5}$ ppmv, and the O$_3$ concentration was set to 30 ppbv. These low NO$_x$ boundary conditions likely caused the perimeter sensitivities to VOC although it should also be noted that the absolute values of these sensitivities are small. The effect of boundary conditions is further discussed later in Section 4.3."**

Minor comments on text:

p. 1, Line 32: The 250 to 50 hPa layer is actually in the stratosphere, so not "stratospheric intrusions"

**Reply:**

**We have revised "stratospheric intrusions" to "stratospheric air mass".**

p. 4, line 4: typo p. 6, line 1: It is not clear to me how the sensitivity to stratospheric O3 is being calculated.

**Reply:**

**Section 2 mentioned our Part 1 paper, and we developed the air mass characterization technique in Part 1 paper. In contrast to this part 1 paper, this part 2 paper directly estimated the sensitivity to stratospheric O$_3$. In this version of CMAQ modeling system, the stratospheric O$_3$ is calculated using O$_3$/PV relationships, and DDM is applied to estimate the sensitivity to this stratospheric O$_3$. To address this comment, we have revised and added the explanation of DDM calculation of stratospheric O$_3$ as follows in Section 3.**

**"In addition, DDM was extended to examine the sensitivity of O$_3$ mixing ratios towards stratospheric O$_3$. A dynamic O$_3$/PV function considering the seasonal, latitudinal and**

**altitude dependencies is constructed at three vertical levels of 58, 76, and 95 hPa fitted as a 5th order polynomial function, and applicable between the range of 50 and 100 hPa (Xing et al., 2016). The sensitivity to this stratospheric $O_3$ is calculated by differentiating the equations used to introduce stratospheric $O_3$ through potential vorticity in the same matter as all other DDM sensitivity calculations. When a user specifies the desire to know the PV sensitivity, a sensitivity field corresponding to the calculation is initialized at the beginning of the model run and then updated with the derivatives in each time step and location where PV calculations occur (typically the uppermost two layers in the model). Since PV ozone in CMAQ is essentially a "replacement" of the ozone field in the top layers before the PV calculations by a scaling function, the same replacement is applied to the first-order sensitivity field. Note that the higher-order sensitivity to this stratospheric $O_3$ is not calculated. This sensitivity is hereafter referred to as O3VORT."**

p. 6, line 10: It should be noted that because the coarse grid resolution smears out NOx, you may be missing locations where O3 is actually VOC-limited, such as urban cores with intense NOx emissions at subgrid scales.

**Reply:**

**We have added the note to explicitly mention the limitation of this coarse-grid analysis as follows:**

**"Note that due to the use of a coarse horizontal grid resolution to cover the entire northern hemisphere, the simulation may not adequately capture the chemical regime in urban areas where O3 chemistry is VOC sensitive."**

**In addition, we have added the statement referring to our previous study as follows:**

**"Due to the coarse grid resolution, H-CMAQ could partly missed the VOC sensitive regime characterized over urban areas, and our previous study reported the dependency of photochemical indicators to judge the $O_3$ regime (e.g., $H_2O_2/(O_3+NO_2)$) on model grid resolution (Zhang et al., 2009)."**

p. 6, lines 21-22: It would be more appropriate to say: "… it can be concluded that ozone is more sensitive to NOx emissions than to biogenic VOCs emissions during April 2010." Also, at some point you should note that not all NOx is anthropogenic (e.g., lightning, soils).

**Reply:**

**We have revised this sentence according to the reviewer's comment.**

p. 10, line 2: Have you identified evidence of "active convection" in the meteorological model, or is this mere speculation?

**Reply:**

**This was our speculation, hence we have revised this sentence as follows:**

**"the latter may be related to active convection"**

p. 12, line 29: It is difficult to follow where results are being presented on a MD8O3 or 24-hour average basis. Those sensitivities can be quite different.

**Reply:**

**MD8O3 is not presented in this study, hence we mentioned the analysis method in the concluding section. We have revised "with other metrics (e.g., MD8O3)" into "with other metrics (e.g., MD8O3) not analyzed here".**

Specific comments on figures:

Fig 1: In the caption, clarify if these are for 8-hour maximum or daily-average results.

**Reply:**

**As we have stated in main caption, these sensitivities are monthly means computed from hourly output. We also explicitly mentioned this point in the figure caption as follows.**

**"The sensitivity coefficients are monthly means computed from all hourly data in April 2010."**

Fig 9 and 10: Caption needs to say what the emissions changes were.

**Reply:**

**We have revised the caption to explicitly explain these emissions changes as follows.**

**For figure 9:**

**"Perspective of changes in $O_3$ concentration resulting from estimated 2010-2015 emission changes over (top panel) U.S.A. and (bottom panel) East Asia at surface, bottom of free troposphere (750 hPa), middle of free troposphere (500 hPa), and top of free troposphere (250 hPa) from left to right."**

**For figure 10:**

**"Perspective of daily and monthly averaged changes in $O_3$ mixing ratio resulting from estimated 2010-2015 emission changes over U.S.A. (light blue bars) and East Asia (light red bars) summarized over four time zones of Pacific, Mountain, Central, and Eastern Standard Time (PST, MST, CST, and EST) in U.S.A."**

---

## Author Comment (AC2) · 17 Dec 2019

Response to Referee Comment 2 by Anonymous Referee #2

Itahashi et al. (2019) investigated the impacts of emissions from East Asia and US on surface O3 over Northern Hemisphere using HDDM in H-CMAQ. They found comparable impacts by the emissions from East Asia and US on surface ozone over western US with US domestic emissions having larger impacts on surface ozone over eastern US while emissions from East Asia have much larger impacts on free troposphere through trans-pacific transport. But they also found the impacts of recent emission changes in East Asia on US O3 levels were small. The manuscript is in general well written but there are several concerns about the methodology used in the manuscript.

**Reply:**

**We thank the reviewer for providing helpful and constructive comments. We have revised our manuscript according to the reviewer's comments and suggestions. We believe that these revisions address all points raised by the reviewer. Our point-by-point responses are provided below, and revisions are indicated in blue in the revised manuscript.**

ZOC of emissions

**We have divided this comment into three smaller parts and replied to each portion individually.**

The estimation on the emission impacts is based on only one H-CMAQ simulation, is that correct? If so, the ZOC estimates are not really accurate as if we reduce all the targeted emissions, meteo/dynamics will also change, which could have feedbacks on the estimations of the emission impacts. The atmospheric conditions will be different under normal and ZOC scenarios. Any explanations on this?

> **Reply:**
>
> **Yes, the estimation is based on one-time simulation of HDDM embedded in CMAQ model. The emission variations can cause the changes in meteorology and dynamics; however, the version of the CMAQ modeling system used in this study is based on the so called "off-line" approach which does not consider the feedback between meteorological variables and chemistry. Therefore, in this study, our estimation just focuses on the emission impacts on the concentration of air pollutants.**

Also, ZOC is not a really realistic scenario and the impacts of ZOC are probably overestimated in this work. So your conclusion about the impacts of the emissions from East Asia may not be that robust. As in your last part of the work, the impacts of recent changes in the emissions from East Asia were found to be small to insignificant. Unless all the emissions from East Asia were removed, significant impacts would exist over western US at surface and entire US in free troposphere? In other

words, To what degree of the changes in the emissions of East Asia should be achieved to show noticeable impacts on US?

**Reply:**

**In this study, HDDM embedded in CMAQ is used to obtain sensitivity coefficients regarding $NO_x$ and NMVOC emissions as shown in Fig. 1. As an example to illustrate the emission impacts, we showed the result of ZOC based on Eq. (5) in Fig. 3. Because the impacts of emissions can be estimated in detail based on Eq. (4), the conclusion can provide the robustness of emission impacts.**

**In the latter parts of the analysis presented in this work, the impacts of recent emission changes are also estimated based on Eq. (4), and these results showed that the impact of emission changes from 2010 to 2015 in East Asia were insignificant on simulated $O_3$ levels over U.S.A. In contrast, as illustrated in Figure 6, a complete removal of emissions from East Asia does result in significant impacts on surface and free-troposphere $O_3$ over U.S.A. can be found if emissions from East Asia is removed. As we already state in Section 4.4, Chinese emissions after 2010 showed complex variations. To assess the possible impacts of these changes on our sensitivity analysis results as well as likely changes in trans-Pacific transported pollution, we summarize the impacts of these emission changes via the illustrations in Figs. 9 and 10.**

For ZOC scenario, does it simply remove all the NOx and NMVOCs emissions? How to distinguish between anthropogenic and natural emissions for NMVOCs? Are the impacts of ZOC the total impacts from the removal of both NOx and NMVOCs? Can we have individual impacts from the removal of NOx only and removal of NMVOCs only? Or maybe we can infer from the sensitivity to these emission? This information could provide more guidance on future emission mitigations.

**Reply:**

**These estimations are based on HDDM embedded in CMAQ. Because of the unified input dataset for H-CMAQ, we first estimated the sensitivities for all emission sources (i.e. anthropogenic and biogenic), and then we further estimated sensitivities to isoprene (a proxy for biogenic emissions) as shown in Fig. S1. HDDM allows both the separate and combined estimation of the impacts of NOx and NMVOC emissions as we have shown in Fig. 1 (a and b for $NO_x$ emissions, c and d for VOCs emissions, and e for the combined effects of $NO_x$ and VOCs ).**

Specific comments:

Page 1, Abstract, line 24, "with a magnitude of about 3 ppbv impacts on a monthly mean …" 3 ppbv O3?

**Reply:**

**Yes, we have added 'O₃' to explicitly mention it.**

Page 4, line 4, "O3 mixing rations and an those of inert tracer …", grammatical error

**Reply:**

**We have corrected this error.**

Page 5, line 5-6, equation (3), should Si,j(2) have the unit of square of the concentration?

**Reply:**

**No, as we have explicitly stated after Eq. (3), the unit of $S_{i,j}^{(2)}$ is same as the concentration, which is ppbv in this case.**

Page 6, and Figure 1, any physical meaning on second-order sensitivity? line 9-10, can you explain more on "concave response"?

**Reply:**

**The second-order sensitivity reflects the nonlinear response. The large value of second-order sensitivity corresponds to the strong nonlinearity, and if the response is linear, the value of second-order sensitivity is negligible. Under the typical $O_3$ concave response to $NO_x$ emissions, the first-order sensitivity is positive and the second-order sensitivity is negative; and the absolute value of this negative second-order sensitivity represents the magnitude of nonlinearity.**

Page 6, line 27-28, what do you mean by "Svocs and Snox"? it is not equal to Svocs +Snox, right? Maybe you mean Svocs-nox ? Please clarify.

**Reply:**

**These equations include two formulae. To avoid the misread, we have revised these equations to be separated by ','.**

Page 7, line 17-18, regions in VOC-sensitive regime are not clearly shown in Fig 2. You may want to change the color scale to improve the quality.

**Reply:**

**The VOC-sensitive regime is not clear as indicated by the weak responses shown in Fig. 1. This is not due to the color scale. We have revised this sentence as follows:**

**"In some areas over the U.S.A. that are characterized by a weak VOC-sensitive regime in Fig. 2…"**

Page 21, Figure 3, are these impacts based on zero-out of both NOx and NMVOCs at the same time? See my general comments above.

**Reply:**

**As we have replied in general comments, these impacts are based on Eq. (5) using the sensitivity coefficients obtained by HDDM. The impacts shown in Figure 3 are estimated by considering both $NO_x$ and VOCs emission changes. HDDM estimates both the individual and combined ozone sensitivities towards these pollutants.**